# Stimulatory Effect of Silver Nanoparticles on the Growth and Flowering of Potted Oriental Lilies

**Piotr Salachna [1,\*]**, **Andżelika Byczyńska [1,\*]**, **Agnieszka Zawadzińska [1]**, **Rafał Piechocki [1]** **and Małgorzata Mizielińska [2]**

1   Department of Horticulture, West Pomeranian University of Technology, 3 Papieża Pawła VI Str., 71-459 Szczecin, Poland; agnieszka.zawadzinska@zut.edu.pl (A.Z.); rafal.piechocki@zut.edu.pl (R.P.)

2   Center of Bioimmobilisation and Innovative Packaging Materials, West Pomeranian University of Technology, 35 Janickiego Str., 71-270 Szczecin, Poland; malgorzata.mizielinska@zut.edu.pl

\*   Correspondence: piotr.salachna@zut.edu.pl (P.S); andzelika.woskowiak@zut.edu.pl (A.B.); Tel.: +48-91-4496-359 (P.S & A.B.)

**Abstract:** Nanoparticles exhibit unique biological activities and may serve as novel plant growth stimulators. This research consisted of a two-year pot experiment designed to find out if silver nanoparticles (AgNPs) might be used in the cultivation of Oriental lilies. In the first year, we evaluated the effects of various concentrations of AgNPs (0, 25, 50, 100, and 150 ppm) and their application methods (pre-planting bulb soaks, foliar sprays, and substrate drenches) on the growth and flowering of *Lilium* cv. Mona Lisa. In the second year, we evaluated the effects of soaking the bulbs of cv. Little John in the same concentration of AgNP solution on plant morphological features, leaf content of photosynthetic pigments, basic macronutrients, and complex biomolecules with the use of the Fourier-transform infrared spectroscopy (FTIR). Soaking the bulbs in a nanoparticle solution turned out to be the most effective strategy for growth and flowering promotion. AgNPs stimulated plant growth, as manifested by enhanced accumulation of leaf and bulb biomass and accelerated flowering. Moreover, plants treated with silver nanoparticles showed higher leaf greenness index, formed more flowers, and flowered longer. At 100 ppm AgNPs, the leaves accumulated the highest content of chlorophyll a, chlorophyll b, and carotenoids, and were the richest in potassium, calcium, and sulfur. The FTIR spectra did not show any changes in absorbance intensity and chemical composition in the leaves from AgNP-treated bulbs.

**Keywords:** *Lilium*; nanosilver; biostimulators; ornamentals; pigments; nutrients; biomacromolecule

## 1. Introduction

Advances in nanotechnology have allowed for the production of specific nanoparticles with unique properties and a wide spectrum of practical applications [1]. Nanoparticles are characterized by small size, low weight, and a high surface to volume ratio [2]. In the agriculture industry, they are increasingly being used as components of new fertilizers [3], plant protection products [4], herbicides [5], and preparations for prolonging cut flower durability [6]. Recently, nanoparticles and nanomaterials have been suggested as potential biostimulators that might improve plant propagation and growth [7,8] and improve plant resistance to stress [9,10]. Using nanoparticles could bring numerous benefits to agriculture and horticulture, but also involves some risks related to their not yet fully recognized environmental impacts [11].

The most interesting metal nanoparticles seem to be silver nanoparticles (AgNPs), which exhibit strong biological activity [12]. They affect plants at many different levels [13,14]. Positive effects of AgNPs include stimulation of germination [15], growth invigoration [16], increased biomass

accumulation [17], improved shoot induction and proliferation [7], or enhanced pigment content [18]. Silver nanoparticles may also show phytotoxicity, manifested by limited germination and seedling growth [19], decreased biomass of leaves and shoots [20], delay in flowering time, and inhibition of photosynthesis [21]. Therefore, further studies are needed to clarify these contradictory observations and to explain the mechanisms controlling growth stimulation or inhibition in AgNP-treated plants.

Production of potted ornamental plants is a rapidly developing area of the horticultural industry [22]. Plants with decorative flowers account for over 50% of potted plants on the market. One such plant is lily (*Lilium* L., Liliaceae), a globally economically important bulbous flower crop. By adding new cultivars every year, lilies are constantly growing in popularity [23]. The most attractive cultivars include oriental hybrids with large, scented flowers. These are also the most difficult to grow, mostly due to their high sensitivity to fungal pathogens [24,25]. The decorative value of ornamental plants may be enhanced by using growth stimulators [26,27]. However, there have been few reports on the effect of biostimulators on the yield of flowers and bulbs in lily [28,29].

This is the first study investigating the effects of different AgNP concentrations and methods of their application on the growth and flowering of potted Oriental hybrid lilies. To better understand plant response to AgNPs, we evaluated the leaf content of assimilation pigments and macronutrients and analyzed macromolecule composition using Fourier-transform infrared spectroscopy (FTIR). With the exception of our earlier research [30], no comprehensive studies have been undertaken on using metal nanoparticles as biostimulators to enhance the ornamental quality of bulbous crop plants.

## 2. Materials and Methods

### 2.1. Plant Material and Growth Conditions

The experiment was carried out between 17 March and 25 July 2015 and between 30 March and 8 August 2016 in an unheated stand-alone tunnel with area of 225 m$^2$, covered with a double layer of plastic with a UV filter, located at the premises of West Pomeranian University of Technology in Szczecin (53°25' N, 14°32' E; 25 m ASL. In the first year, we investigated cv. Mona Lisa, and in the second, cv. Little John. Both cultivars belong to a division of Oriental hybrids and are recommended for pot cultivation. Bulbs 12–13 cm in circumference were imported from Dutch breeding plantations and kept in cold storage at 6–8 °C prior to planting. The experiments included only healthy bulbs, with no spots or damage and with strong and unbroken roots.

Each year the bulbs were planted individually into 16 cm diameter round plastic pots with a volume of 2 dm$^3$. The pots were filled with TS1 peat substrate (Klasmann-Delimann, Germany), with a pH in H$_2$O 5.4 and salinity of 1.09 g NaCl dm$^{-3}$, containing 162 mg dm$^{-3}$ N-NO$_3$, 118 mg dm$^{-3}$ P, 323 mg dm$^{-3}$ K, 1189 mg dm$^{-3}$ Ca, 132 mg dm$^{-3}$ Mg, and 17 mg dm$^{-3}$ Cl. The plants were grown on steel tables with a density of 16 plants per m$^2$ under a natural photoperiod, and mean air temperature in the tunnel (2015/2016) was: March 9.9/9.0 °C, April 11.9/12.4 °C, May 15.4/19.8 °C, June 18.1/22.0 °C, and July 21.4/21.3 °C.

### 2.2. Treatments

AgNPs purchased from Sigma Aldrich (particle size <100 nm and surface area 5.0 m$^2$ g$^{-1}$) were dissolved in deionized water and used at the following concentrations: 0 (control), 25, 50, 100, and 150 ppm. In the first year of the study, three application methods were investigated: (I) bulb soaking in AgNP solution for 60 min prior to planting, (II) substrate drenches (100 mL/pot), and (III) foliar sprays (45 mL/plant). The plants were drenched or sprayed with AgNP solutions three times, i.e., 30, 40, and 50 days after planting. In the second year, AgNPs were applied only in the form of bulb soaking prior to planting. Both experiments were arranged in a randomized complete block design, and each treatment was replicated four times with 20 bulbs per treatment.

### 2.3. Growth and Flowering Characteristics

We determined daily the number of days from bulb planting to the beginning of anthesis and flower longevity. The beginning of anthesis was assumed as the moment when one flower per plant was fully opened. Flower longevity meant the number of days from the beginning of anthesis to the opening of the last flower. The parameters measured at the beginning of anthesis included: plant height from the soil line to the uppermost part of inflorescence, number of leaves per plant, tepal length, and tepal width. Additionally, the SPAD (soil and plant analysis development) leaf greenness index was measured with a Chlorophyll Meter SPAD 502 (Minolta, Osaka, Japan) in three fully developed leaves per plant by taking three readings per leaf and calculating mean values. When the flowering ceased, we determined the number of flowers per inflorescence, fresh weight of leaves and bulbs, and the number of scales per bulb. In the second year of the experiment, we analyzed leaves harvested from the central section of the stem of three representative plants of each treatment.

### 2.4. Chlorophylls and Carotenoids

Pigment content was determined spectrophotometrically in fresh leaves (10 g fresh weight). To this end, discs of the same diameter (7 mm) were cut out with a cork borer from the central part of the leaf. The pigments were extracted with 99% N-N-dimethylformamide for 24 h. Chlorophyll a, chlorophyll b, chlorophyll a + b, and carotenoid presence were detected by reading their absorbance at 440, 645, and 663 nm with a spectrophotometer SPEKOL 11 (Carl Zeiss Jena, Jena, Germany). The pigment content was calculated using the appropriate formulae [31,32] and has been expressed in mg kg$^{-1}$ fresh weight (FW).

### 2.5. Macronutrient Concentration

Fresh leaves (100 g fresh weight) were rinsed thrice with distilled water, dried at 60 °C to dry weight, and ground. Leaf tissue samples were then microwave digested in $HNO_3$, using closed Teflon vessels. Phosphorus (P), potassium (K), magnesium (Mg), calcium (Ca), and sulfur (S–$SO_4$) were determined using inductively coupled plasma–optical emission spectrometry ICP-OES (Optima 2000TM DV PerkinElmer, Waltham, MA, USA). Nitrogen (N) content was established with a Kjeldahl apparatus (Vapodest, Gerhardt, Germany). Leaf macronutrient content has been expressed in % of dry weight (DW).

### 2.6. Fourier-Transform Infrared Spectroscopy (FTIR) Analysis

Fourier-transform infrared (FTIR) spectrum of dry leaf tissue (2 g) was measured using a FTIR spectroscope (PerkinElmer Spectrophotometer, Spectrum 100), operated at a resolution of 4 cm$^{-1}$ for four scans. Samples were ground into powder and placed directly (each one separately) at the ray-exposing stage. The spectrum recorded a wave number of 650–4000 cm$^{-1}$.

### 2.7. Experimental Design and Statistical Analysis

The experiment tested two factors in the first year (five concentrations × three application methods) and a single factor in the second year (five concentrations). Each biometric measurement included three similar plants from each repetition ($n = 12$). The plant material was analyzed in three independent biological replicates ($n = 3$). The results were subjected to ANOVA using Statistica Professional 13.3 package (TIBCO Software, Palo Alto, CA, USA). Tukey's test at $p \leq 0.05$ was used to assess the smallest significant differences between means.

## 3. Results

### 3.1. Impact of AgNPs on Plant Growth and Flowering

The outcomes from the first year indicated positive effects of AgNPs on the majority of the assessed morphological parameters in lily cv. Mona Lisa (Figure 1, Tables 1 and 2). All plants treated with AgNPs clearly showed enhanced leaf fresh weight (by 23.6–50.5%), greenness index (by 8.6–18.4%), and bulb fresh weight (by 44.9–73.4%), and began flowering by two or three days earlier than control plants. At 50 ppm AgNPs, the lilies grew the tallest (48.1 cm), produced the greatest number of leaves (27.1), and flowered for the longest time (11.4 days), and their bulbs had the greatest fresh weight (37.1 g), and formed the greatest number of scales (23.0). The tested AgNP concentrations and application methods did not significantly affect tepal length or width (Table 2).

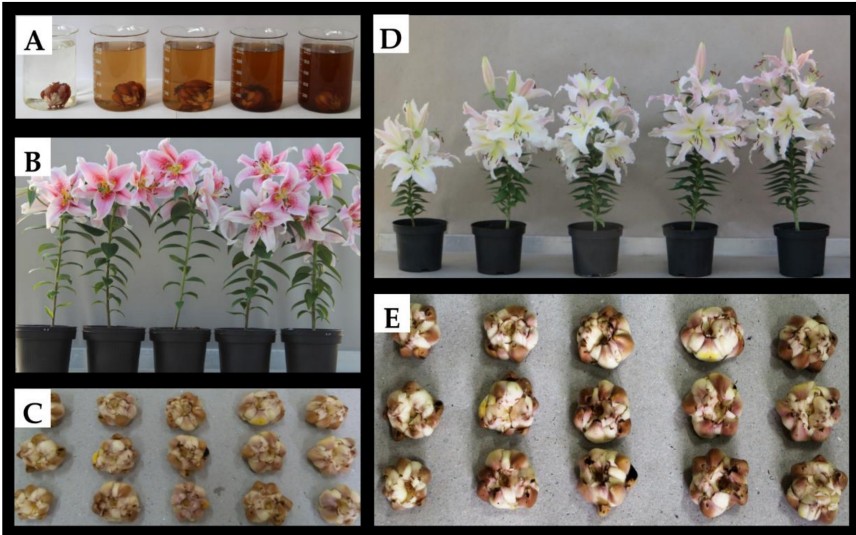

**Figure 1.** Effects of silver nanoparticles (AgNPs) on color change of solution (**A**); flowering (**B**) and bulb yield (**C**) of lily cv. Mona Lisa; flowering (**D**) and bulb yield (**E**) of lily cv. Little John after the application of AgNP pre-planting bulb soaks. Left to right: nontreated control, 25, 50, 100, and 150 ppm AgNPs.

**Table 1.** Main effects of silver nanoparticle (AgNP) concentration and application method on growth of lily cv. Mona Lisa. Values followed by differing letters in each column are significantly different at $p \leq 0.05$ (ANOVA and Tukey's test). *, **, ***: significant at $p \leq 0.05$, 0.01, or 0.001, respectively. Non-significant: ns.

| AgNP Treatment | Plant Height (cm) | No. of Leaves Per Plant | Greenness Index (SPAD) | Leaf Fresh Weight (g/Plant) | Bulb Fresh Weight (g/Plant) | No. of Scales Per Bulb |
|---|---|---|---|---|---|---|
| Concentration (C) | | | | | | |
| 0 ppm | 44.7 b | 21.3 c | 52.3 c | 20.8 c | 21.4 c | 18.5 c |
| 25 ppm | 45.7 ab | 25.6 ab | 56.8 b | 27.3 a | 31.3 b | 19.8 bc |
| 50 ppm | 48.1 a | 27.1 a | 61.5 a | 28.1 a | 37.1 a | 23.0 a |
| 100 ppm | 46.8 ab | 23.5 bc | 61.9 a | 31.3 a | 35.1 a | 22.9 a |
| 150 ppm | 45.6 ab | 23.3 bc | 61.2 a | 25.7 b | 31.0 b | 21.1. ab |
| Method (M) | | | | | | |
| Bulb soaks | 47.2 a | 26.9 a | 61.2 a | 28.2 a | 33.3 a | 23.2 a |
| Drenches | 45.0 b | 22.4 b | 58.1 b | 26.1 b | 31.0 b | 20.3 b |
| Foliar sprays | 46.4 ab | 23.2 b | 57.0 b | 25.6 b | 29.2 c | 19.8 b |
| Two-way ANOVA | | | | | | |
| C | ** | *** | *** | *** | *** | ** |
| M | * | *** | ** | ** | *** | *** |
| C × M | ns | ** | * | ns | *** | ns |

**Table 2.** Main effects of silver nanoparticle (AgNP) concentration and application method on flowering of lily cv. Mona Lisa. Values followed by differing letters in each column are significantly different at $p \le 0.05$ (ANOVA and Tukey's test). *, **: Significant at $p \le 0.05$, 0.01 or 0.001, respectively. Non-significant: ns.

| AgNP Treatment | Days to Anthesis | No. of Flowers Per Plant | Tepal Length (cm) | Tepal Width (cm) | Flower Longevity (Days) |
|---|---|---|---|---|---|
| Concentration (C) | | | | | |
| 0 ppm | 109 b | 2.7 b | 11.7 a | 5.76 a | 9.50 b |
| 25 ppm | 107 a | 3.6 ab | 11.9 a | 5.76 a | 10.3 ab |
| 50 ppm | 106 a | 4.0 a | 12.0 a | 5.89 a | 11.4 a |
| 100 ppm | 106 a | 4.0 a | 12.0 a | 5.93 a | 11.0 ab |
| 150 ppm | 107 a | 3.4 ab | 12.0 a | 5.80 a | 10.6 ab |
| Method (M) | | | | | |
| Bulb soaks | 106 a | 4.0 a | 11.9 a | 5.91 a | 10.3 a |
| Drenches | 107 b | 3.1 b | 11.8 a | 5.74 a | 11.0 a |
| Foliar sprays | 107 b | 3.4 b | 12.0 a | 5.84 a | 10.4 a |
| Two-way ANOVA | | | | | |
| C | ** | * | ns | ns | * |
| M | * | ** | ns | ns | ns |
| C × M | ns | ns | ns | ns | ns |

As for the AgNP application methods, cv. Mona Lisa lilies grown from the bulbs soaked in nanoparticle solutions produced significantly more leaves and flowers and had higher greenness indices and fresh weight of leaves and bulbs than plants watered or sprayed with AgNPs. In addition, soaking the bulbs in AgNP solution accelerated flowering and increased the number of flowers without affecting flower longevity (Tables 1 and 2).

For the number of leaves, greenness index, and bulb fresh weight, we found a significant interaction between AgNP concentration and its application method (Table 3).

**Table 3.** The interaction effects of application method and silver nanoparticle (AgNP) concentration on the number of leaves, greenness index, and bulb fresh weight of lily cv. Mona Lisa. Values followed by differing letters in each column are significantly different at $p \le 0.05$ (ANOVA and Tukey's test).

| AgNP Treatment | | No. of Leaves Per Plant | Greenness Index (SPAD) | Bulb Fresh Weight (g/Plant) |
|---|---|---|---|---|
| Method | Concentration | | | |
| Bulb soaks | 0 ppm | 21.3 c | 52.3 d | 21.4 g |
| | 25 ppm | 29.2 ab | 59.2 abcd | 40.2 a |
| | 50 ppm | 24.8 bc | 68.3 a | 39.3 ab |
| | 100 ppm | 24.5 bc | 66.1 ab | 33.4 cd |
| | 150 ppm | 34.5 a | 60.1 abcd | 32.2 de |
| Drenches | 0 ppm | 21.3 c | 52.3 d | 21.4 g |
| | 25 ppm | 24.0 bc | 56.1 cd | 26.4 fg |
| | 50 ppm | 22.0 c | 56.7 cd | 40.9 a |
| | 100 ppm | 23.0 bc | 62.1 abc | 33.8 bcd |
| | 150 ppm | 21.8 c | 63.3 abc | 32.6 cde |
| Foliar sprays | 0 ppm | 21.3 c | 52.3 d | 21.4 g |
| | 25 ppm | 23.7 bc | 55.1 cd | 27.4 ef |
| | 50 ppm | 23.1 bc | 59.5 abcd | 31.0 def |
| | 100 ppm | 23.0 bc | 57.6 bcd | 37.9 abc |
| | 150 ppm | 25.1 bc | 60.3 abcd | 28.4 def |

In the second year of the study, we focused on cv. Little John and chose the most effective method of AgNP application, i.e., bulb soaking. The data we obtained for this cultivar confirmed the stimulatory effects of AgNPs on lily growth and flowering (Figure 1, Table 4). Plants treated with all

investigated concentrations of AgNPs reached significantly greater fresh weight of leaves (28.6–47.9%) and bulbs (40.6–56.5%), and produced more flowers (4.5–14.9%) with longer tepals (9.7–13.7%) than control ones. Moreover, AgNP presence accelerated flowering by two to four days, except for the concentration of 150 ppm. The most beneficial AgNP concentration was 100 ppm, at which the plants were the highest (56.5 cm), produced the greatest number of leaves (65.7), had the highest greenness index (59.4 SPAD), the longest (14.1 cm) and the widest (7.03 cm) tepals, and flowered for the longest time (18.2 days).

**Table 4.** Effects of silver nanoparticle (AgNP) pre-planting bulb soaks on growth and flowering of lily cv. Little John. Values for each parameter followed by differing letters are significantly different at $p \leq 0.05$ (ANOVA and Tukey's test). *, **, ***: Significant at $p \leq 0.05$, 0.01 or 0.001, respectively.

| Parameters | AgNP Concentration | | | | | One-Way ANOVA |
|---|---|---|---|---|---|---|
| | 0 | 25 ppm | 50 ppm | 100 ppm | 150 ppm | |
| Plant height (cm) | 48.8 a | 53.3 b | 54.7 ab | 56.5 a | 53.5 b | ** |
| No. of leaves per plant | 58.3 b | 59.3 ab | 63.0 ab | 65.7 a | 61.7 ab | * |
| Greenness index (SPAD) | 48.5 c | 54.6 b | 56.6 ab | 59.4 a | 55.4 ab | ** |
| Leaves fresh weight (g) | 35.3 b | 45.4 a | 50.6 a | 52.2 a | 50.0 a | ** |
| Bulb fresh weight (g) | 52.4 b | 73.7 a | 79.8 a | 82.0 a | 74.2 a | *** |
| No. of scales per bulb | 31.7 c | 38.3 b | 39.7 b | 43.0 a | 43.3 a | ** |
| Days to anthesis | 113 b | 110 a | 110 a | 109 a | 111 ab | * |
| No. of flowers per plant | 6.7 b | 7.5 a | 7.7 a | 7.7 a | 7.0 a | * |
| Tepal length (cm) | 12.4 b | 13.6 a | 13.7 a | 14.1 a | 13.6 a | ** |
| Tepal width (cm) | 6.19 b | 6.80 ab | 6.84 ab | 7.03 a | 6.93 ab | * |
| Flower longevity (days) | 14.2 b | 15.3 ab | 16.5 ab | 18.2 a | 15.5 ab | * |

*3.2. Effect of AgNPs on Photosynthetic Pigments and Macronutrient Concentration*

Our study showed a significant effect of AgNPs on the leaf content of photosynthetic pigments (Figure 2). Leaves of the lily cv. Little John treated with 100 ppm of AgNPs accumulated the greatest amounts of chlorophyll a, chlorophyll b, chlorophyll a + b, and carotenoids. For this treatment, the levels of chlorophyll a, chlorophyll b, chlorophyll a + b, and carotenoids were higher by 31.2, 23.1, 28.6, and 26.3% than in control, respectively. Leaf content of nitrogen, potassium, calcium, and sulfur also depended on AgNP concentration (Figure 3). In comparison with the control plants, AgNPs at 50 ppm enhanced nitrogen and potassium content by 9.2 and 16.1%, respectively. Plants treated with 100 ppm AgNPs also showed significantly higher levels of potassium (by 14.9%), as well as calcium (by 14.4%) and sulfur (by 25.5%). The contents of phosphorus and magnesium were unaffected by AgNP treatments (Figure 3).

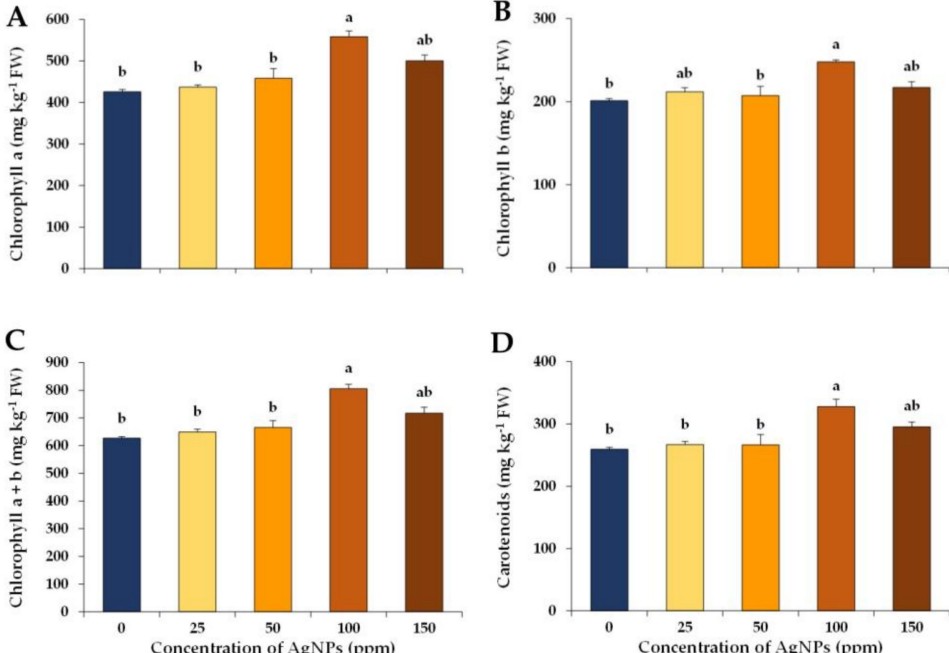

**Figure 2.** Effects of silver nanoparticle (AgNP) pre-planting bulb soaks on chlorophyll a (**A**), chlorophyll b (**B**), chlorophyll a + b (**C**), and carotenoid (**D**) content of lily cv. Little John. Vertical bars indicate the standard error (SE) of the mean. A different lower-case letter above each bar in each panel indicates a significant difference between treatment at $p \leq 0.05$ (ANOVA and LSD test). FW: fresh weight.

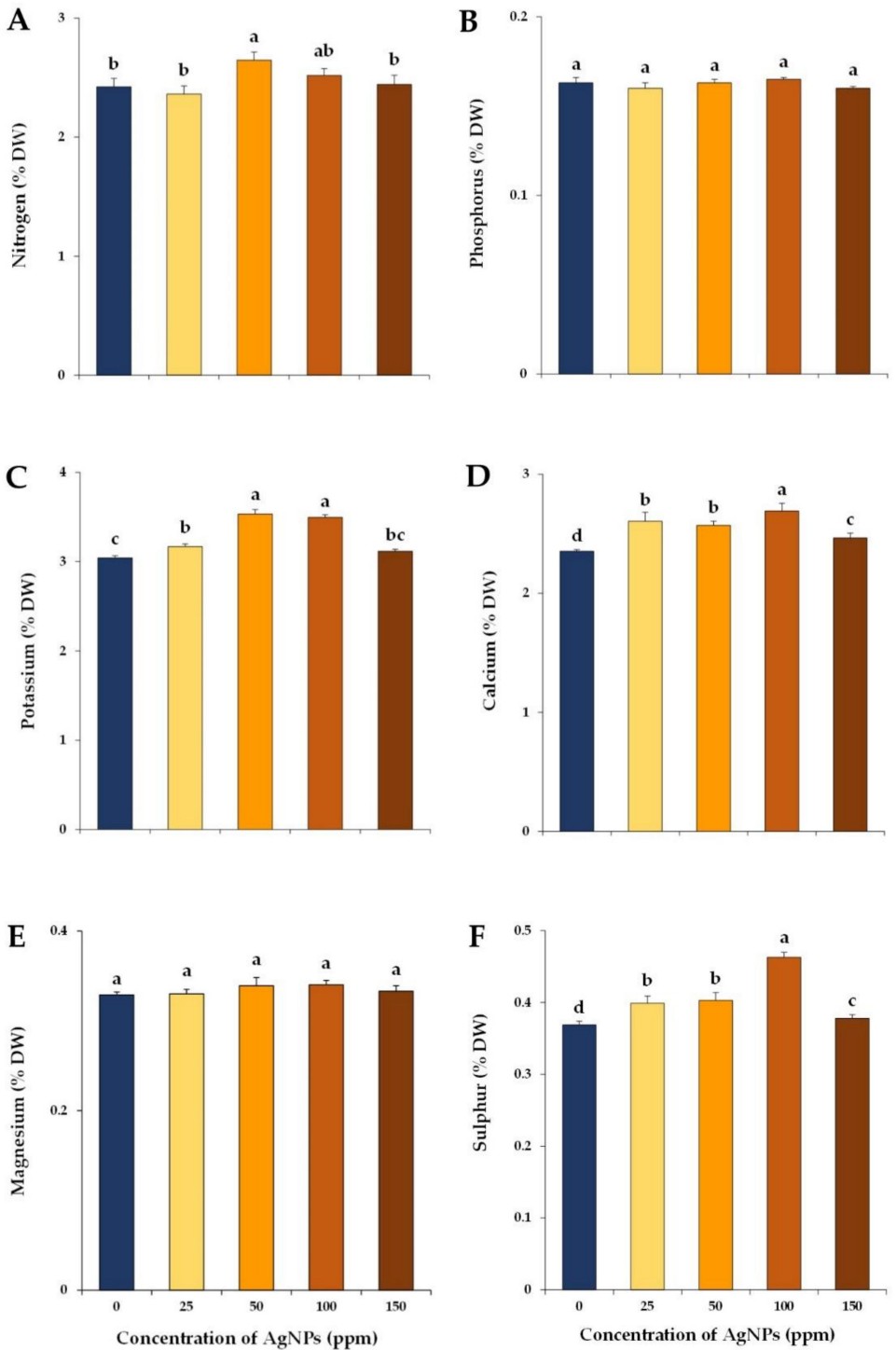

**Figure 3.** Effects of silver nanoparticles (AgNPs) preplant bulb soaks on leaf nitrogen (**A**); phosphorus (**B**); potassium (**C**); calcium (**D**); magnesium (**E**) and sulphur (**F**) content of lily cv. Little John. Vertical bars indicate the standard error (SE) of the mean. A different lower-case letter above each bar in each panel indicates a significant difference between treatment at $p \leq 0.05$ (ANOVA and LSD test). DW: dry weight.

### 3.3. FTIR Analysis

The study analyzed five regions in the FTIR spectra: (1) from 3600 to 3200 cm$^{-1}$, (2) from 3200 to 2800 cm$^{-1}$, (3) from 1800 to 1500 cm$^{-1}$, (4) from 1400 to 1200 cm$^{-1}$, and (5) from 1200 to 900 cm$^{-1}$

(Figure 4). In the case of the 3285.90 cm$^{-1}$ peak, its consistency with absorption was noted, stimulated by O–H single bonds. By contrast, spectrum peaks at 2917.58, 2850.42, and 1411.25 cm$^{-1}$ were consistently observed for a peak with CH$_3$–CH$_2$ induced absorption. The 1590.73 cm$^{-1}$ and 1587.03 cm$^{-1}$ peaks corresponded to lipids and pectin. Spectrum peaks at 1026.40, 1026.85, and 1098.49 cm$^{-1}$ were consistently observed for a peak with O–H- and C–O-induced absorption, which mainly occurs in carbohydrates [33]. The FTIR analysis did not show any influence of AgNPs on the location and relative intensity of the oscillation bands.

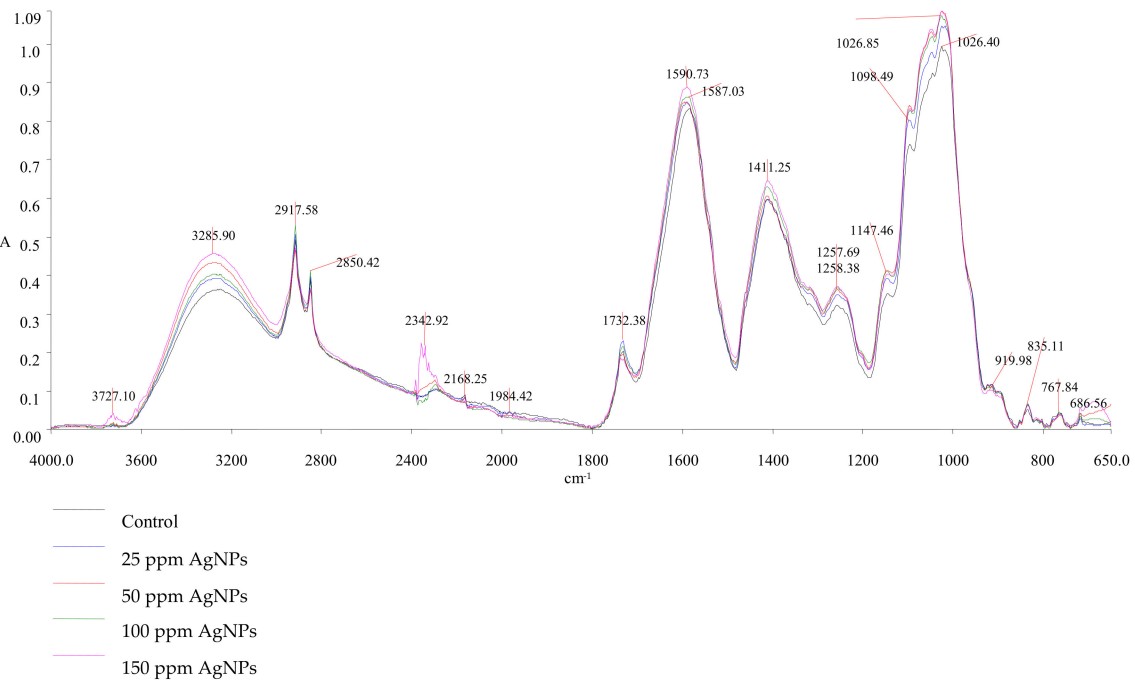

**Figure 4.** Fourier-transform infrared (FTIR) spectra (region 4000–650 cm$^{-1}$) from leaf samples of lily cv. Little John after the application of silver nanoparticle (AgNP) pre-planting bulb soaks.

## 4. Discussion

The study demonstrated positive effects of AgNPs on lily production, as manifested in enhanced growth and more abundant flowering. Plants treated with AgNPs responded with accelerated anthesis, higher greenness index, greater number of flowers, and prolonged flowering, which translated into a greater decorative and commercial value than that of non-treated plants. Additionally, AgNPs considerably increased bulb weight and the number of scales, reflecting improved reproduction potential of the species. Leaves of lilies treated with AgNPs showed no signs of necrosis, chlorosis, leaf and bud drying, or other symptoms of decreased plant quality (Figure 1). Lily cv. Mona Lisa developed the most preferable morphological features when treated with 50 ppm AgNPs, while for cv. Little John, the optimal concentration of nanoparticles was 100 ppm. We have previously reported similar results confirming the beneficial effects of 100 ppm AgNPs, applied by pre-planting soak, on the yield of cut flowers and daughter bulbs of *Tulipa gesneriana* cv. Pink Impression [30]. Dipping cuttings in 50 and 100 ppm AgNP solutions in lily propagation via bulb scales increased the weight of bulblets and their adventitious roots [34]. In in vitro cultures of lily, supplementation of media with AgNPs stimulated morphogenesis and elimination of bacterial contaminations [35]. Information on using AgNPs in the cultivation and propagation of ornamental bulbous plants is scarce, and while data on AgNP treatment in other plant groups are more abundant, they are often contradictory. Phytostimulatory effects of biosynthesized AgNPs on seedling elongation and their biomass growth have been reported in *Oryza sativa* [17] and *Trigonella foenum-graceum* [16]. Treatment

of *Chrysanthemum morifolium* with 7.5 ppm AgNPs applied to tissue cultures significantly improved plant height, length and width of leaves, and plant fresh and dry weight [36]. In *Swertia chirata*, supplementation of the cultivation media with AgNPs of 20 nm in diameter favorably influenced shoot morphogenesis [7]. Contrary to that, exposing *Capsicum annum* to soil-drench-applied AgNPs inhibited plant growth and reduced biomass accumulation of the above-ground parts [20]. Seedlings of *Lupinus termis* treated with high concentrations of AgNPs (300–500 ppm) responded with a reduction in shoot and root elongation and decreased biomass accumulation. However, AgNPs at 100 ppm stimulated shoot and root growth [37]. Similarly, positive effects of low-concentrated solution of AgNP on seedling development were reported in *Pisum sativum* [38]. Inconsistent plant responses to AgNPs may be due to the fact that nanoparticle actions depend on plant genotype [39,40], AgNP concentration [41,42], and application method [43], as confirmed in our study. Bulb soaking in various concentrations of nanoparticles turned out to be more effective than drenching or spraying. It could be assumed that bulbs treated with AgNPs prior to planting were more resistant to soil pathogen infections, and thus their growth and development was more vigorous from the beginning of the cultivation. AgNPs are known for their high antimicrobial activity and effective plant protection against various diseases [1,4,12]. A suggested beneficial effect on lily disease of AgNPs still needs to be further studied. Other factors, such as nanoparticle size, shape [44], or synthesis method [41,45] also make it difficult to compare the results of different studies and to explain the mechanisms of AgNP action. Different plant responses to different AgNP doses may be the effect of hormesis, i.e., a stimulatory influence of low doses and inhibitory influence of high doses of the same agent [46]. According to Juárez-Maldonado et al. [47], the surface charges of nanoparticles interact with the surface charges of plant cells, inducing plant responses from biostimulation to toxicity. These authors assume a two-stage biostimulation process. The initial stage of physicochemical character involves the interaction of surface charges, while the later stage consists of a series of biochemical stimuli that promote nanoparticle penetration into the cells or cause changes in the membranes or integral proteins [47].

Leaf contents of assimilation pigments and macronutrients serve as important biomarkers of plant physiological condition. Increased synthesis of chlorophyll and more effective uptake of minerals are known to considerably improve plant growth and enhance biomass production. In our study, pre-planting bulb soaks of AgNPs at 100 ppm increased leaf content of chlorophyll a, chlorophyll b, and carotenoids, as well as the levels of potassium, calcium, and sulfur. We also saw enhanced accumulation of potassium, accompanied by greater content of nitrogen in plants treated with AgNPs at 50 ppm. Stimulation of chlorophyll synthesis following AgNP foliar application has been demonstrated previously in the leaves of *Triticum aestivum* [18]. AgNPs are assumed to exert a pleiotropic effect on plants by affecting their physiological and biochemical processes and gene expression profiles. Supplementing soil substrate with AgNPs resulted in considerable increase in leaf chlorophyll content, nitrogen and phosphorus uptake, accumulation of crude protein, and enhanced expression of mRNA for nitrate reductase and ferredoxin in a culture of *Phaseolus vulgaris* [41]. Syu et al. [44] demonstrated that phytostimulatory activity of medium-applied AgNPs on the growth of *Arabidopsis* correlated with accumulation of proteins associated with the cell cycle and carbohydrate metabolism, and changes in the expression of the genes involved in multiple cellular processes, such as cell proliferation, photosynthesis, and signaling pathways of such hormones as auxins, abscisic acid, and ethylene. Gupta et al. [17] reported AgNP-stimulated growth improvement in *Oryza sativa* seedlings accompanied by elevated levels of catalase, ascorbate peroxidase, and glutathione reductase, and decreased amounts of lipid peroxidation and hydrogen peroxide content. By enhancing the activity of antioxidant enzymes, the nanoparticles probably help to reduce oxidative stress and may reinforce plant responses to other types of stresses, such as salinity or high temperature [48,49].

Macromolecules are the main building blocks of plant bodies and their endogenously determined composition may be modified by external factors, including nanoparticles and nanocompounds [50]. To obtain comprehensive data on the composition of plant macromolecules, we carried out vibrational spectroscopy FTIR that enabled the identification of individual functional groups and detection of

changes in various chemical bonds. We found that AgNP treatment did not alter the macromolecular composition of lily leaves. However, our findings were inconsistent with a study by Zuverza-Mena et al. [33], where FTIR analysis detected changes in the bands corresponding to lipids (3000–2800 cm$^{-1}$), proteins (1550–1530 cm$^{-1}$), and structural components such as lignin, pectin, and cellulose in *Raphanus sativus* seedlings grown in AgNP suspensions. In *Lycopersicum esculentum*, grown in sewage sludge amendment soil containing TiO$_2$ nanoparticles, FTIR analysis revealed a decrease in tannins and lignins and an increase in carbohydrates in leaves, but no changes in fruits [51]. The effect of AgNPs on the physiological responses, mineral status, and macromolecule conformation in plants is still an open question.

## 5. Conclusions

Our study showed that AgNPs used in the form of bulb-soaking solutions stimulated growth and flowering of two cultivars of Oriental lily. AgNP application resulted in enhanced leaf and bulb biomass, leaf greenness index, and flower abundance. Additionally, the plants treated with AgNPs began their anthesis earlier and featured prolonged flower longevity, meaning their decorative period was longer, which is a top priority in floriculture. The effects of AgNPs on plant growth and the content of assimilation pigments and some macronutrients depended on nanoparticle concentration. In most cases, the best effects were achieved for 50 and 100 ppm. FTIR spectroscopy showed no quantitative or qualitative changes in macromolecules and individual functional groups in response to AgNP treatment. The unique properties of AgNPs may be highly beneficial in the cultivation of ornamental bulb plants, but, as their mechanisms of action are not fully understood, further detailed mycological, biochemical, and molecular studies on the impact of nanosilver on plant health and stress are necessary.

**Author Contributions:** Conceived and designed the experiments, A.B., P.S. and A.Z.; Collected and analyzed the data, A.B., M.M. and R.P., Wrote the paper: P.S., A.B. and M.M.

**Funding:** The study was supported by the Polish Ministry of Science and Higher Education (Project UPB 517-07-014-5365/17 ZUT).

**Acknowledgments:** The authors would like to thank Róża Stuart for her help in the lab.

**Conflicts of Interest:** The authors declare no conflict of interest.

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
