# Peer review of "Stimulatory Effect of Silver Nanoparticles on the Growth and Flowering of Potted Oriental Lilies"

_agronomy, doi:10.3390/agronomy9100610_

Round 1

Reviewer 1 Report

The language needs to be tightened in places. It appears Ref# 44 is not cited in the narrative. Although this is an interesting report, there is no explanation as to how nanoparticle treatment modifies plant development to account for differences observed among treatments. Effort should at least have been made to to collect data on the effect of AgNP treatments on disease prevalence and oxidative stress; both suggested in the discussion as potential pathways by which nanoparticles may improve plant performance.

Author Response

Author's Reply to the Review Report (Reviewer 1)

Although this is an interesting report, there is no explanation as to how nanoparticle treatment modifies plant development to account for differences observed among treatments. Effort should at least have been made to to collect data on the effect of AgNP treatments on disease prevalence and oxidative stress; both suggested in the discussion as potential pathways by which nanoparticles may improve plant performance.

Response: Thank you for the positive assessment of our manuscript in terms of the introduction, research design, methodology and result presentation.  The review form lacks comments on our conclusions. In our answers to the review, we outlined further research directions that would improve our understanding of nanosilver mechanism of action. We would like to underline that our study focused on the effects of nanosilver on growth and flowering of lily and chemical composition of its leaves. The experiments were conducted for two years in production conditions, so that the results are reliable and actionable in practice. We believe that broader research on the effects of nanosilver on individual pathogenic fungi, as well as oxidative stress evaluated based on changes in enzyme activity or gene expression should be conducted in more controlled conditions (growth chambers, in vitro), providing better control over external factors. 

The language needs to be tightened in places. It appears Ref# 44 is not cited in the narrative.

Response: Done.

Reviewer 2 Report

Report on agronomy-572558

This manuscript describes the effects of concentrations and methods of application of silver nanoparticles on the growth and flowering of lilies; the approach is not so novel, indeed the authors have published something similar on tulip. Nevertheless, it is an interesting set of results, that deserve to be published, but not in the current form. I suggest some moderate revision, but encourage the authors to attend to the issues raised.

The authors will note that I have annotated heavily the manuscript, in the attached pdf file. I recommend that the authors look at every comment, and review, accept, or reject if there is sufficient cause.

Taking the sections in order:

Abstract, some rearrangement in the order of the sentences is needed, to make the abstract flow logically.

Introduction, a sound section, covering the salient literature.

Materials and methods, my queries about numbers of plants and replications are partly answered in the section 2.7, but still some queries are pertinent. And why say 6 concentrations if 5 were used. Was one set of data [for the missing concentration] omitted? Also, how frequent were the measurements of anthesis, if not every day [indicate in text] then how can the difference of one day be significant in Table 2..?

Results, although the authors indicate differences between treatments, some of the differences are not absolute, and some treatments [e.g., lines 173/4, where they say 50 ppm produced bulbs of greatest weight, in fact they did not differ from bulb soaking at 100 and 150 ppm.

Discussion, mentions previously noted effects of AgNP on diseases, but unfortunate no relevant measures were made by them. So, the statement in lines 259-261 is barely justified. We know nothing of the pathogen status of the imported bulbs nor the peat substrate  used. Is it likely that either were infected with soil pathogens? Make reference to the manner of application of AgNP on page 11, so it is more relevant to the actual findings. Check the reference in line 318, incorrect as is.

Conclusions, a little care needed with repetition.

References, a couple of minor [but important] queries.

Figures, take care with units in Figure 2.

Overall, a sound set of results, but ensure not to conclude beyond the conditions [i.e., pathogen effect], of the experiments.

Author Response

Author's Reply to the Review Report (Reviewer 2)

We are deeply grateful for valuable comments that helped us improve the manuscript. Please find our answers below.

Abstract, some rearrangement in the order of the sentences is needed, to make the abstract flow logically.

Response: Abstract was structured and ordered.

Materials and methods, my queries about numbers of plants and replications are partly answered in the section 2.7, but still some queries are pertinent.

Response: We added detailed information on the number of plants per replication.

And why say 6 concentrations if 5 were used. Was one set of data [for the missing concentration] omitted?

Response: We made a mistake in the text mentioning 6 instead of 5 concentrations and fixed it now.

Also, how frequent were the measurements of anthesis, if not every day [indicate in text] then how can the difference of one day be significant in Table 2..?

Response: In the flowering period, we recorded our observations daily. Information on that was provided in the text.

Results, although the authors indicate differences between treatments, some of the differences are not absolute, and some treatments [e.g., lines 173/4, where they say 50 ppm produced bulbs of greatest weight, in fact they did not differ from bulb soaking at 100 and 150 ppm.

Response: To make our results on interactions clear and transparent, we have not discussed individual features in details.

Discussion, mentions previously noted effects of AgNP on diseases, but unfortunate no relevant measures were made by them. So, the statement in lines 259-261 is barely justified. We know nothing of the pathogen status of the imported bulbs nor the peat substrate used. Is it likely that either were infected with soil pathogens? Overall, a sound set of results, but ensure not to conclude beyond the conditions [i.e., pathogen effect], of the experiments.

Response: The section on the effects of nanosilver on plant condition was limited, as the Reviewer rightly pointed out that this was outside the manuscript scope.

Make reference to the manner of application of AgNP on page 11, so it is more relevant to the actual findings.

Response: We supplemented the Discussion with data on methods for nanosilver application.

Check the reference in line 318, incorrect as is.

Response: Done.

Conclusions, a little care needed with repetition.

Response: Done.

References, a couple of minor [but important] queries.

Response: Done.

Figures, take care with units in Figure 2.

Response: Done.